# Salient Targets and Fear of Falling Changed the Gait Pattern and Joint Kinematic of Older Adults

**DOI:** 10.3390/s22239352

**Published:** 2022-12-01

**Authors:** Yue Luo, Xiaojie Lu, Nicolas S. Grimaldi, Sherry Ahrentzen, Boyi Hu

**Affiliations:** 1Department of Industrial and Systems Engineering, University of Florida, Gainesville, FL 32611, USA; 2Shimberg Center for Housing Studies, University of Florida, Gainesville, FL 32611, USA; 3Microelectronics Thrust, Function Hub, The Hong Kong University of Science and Technology (Guangzhou), Guangzhou 511453, China; 4J. Crayton Pruitt Family Department of Biomedical Engineering, University of Florida, Gainesville, FL 32611, USA

**Keywords:** visual cues, fear of falling, older adults, gait, joint kinematics

## Abstract

Background: Fear of falling and environmental barriers in the home are two major factors that cause the incidence of falling. Poor visibility at night is one of the key environmental barriers that contribute to falls among older adult residents. Ensuring their visual perception of the surroundings, therefore, becomes vital to prevent falling injuries. However, there are limited works in the literature investigating the impact of the visibility of the target on older adults’ walking destinations and how that impact differs across them with different levels of fear of falling. Objective: The purpose of the study was to examine the effects of target salience on older adults’ walking performance and investigate whether older adults with varying levels of fear of falling behave differently. Methods: The salient target was constructed with LED strips around the destination of walking. Fifteen older adults (aged 75 years old and above), seven with low fear of falling and eight with high fear of falling, volunteered for the study. Participants walked from the designated origin (i.e., near their beds) to the destination (i.e., near the bathroom entrance), with the target turned on or off around the destination of the walking trials. Spatiotemporal gait variables and lower-body kinematics were recorded by inertial sensors and compared by using analysis of variance methods. Results: Data from inertial sensors showed that a more salient target at the destination increased older adults’ gait speed and improved their walking stability. These changes were accompanied by less hip flexion at heel strikes and toe offs during walking. In addition, older adults with low fear of falling showed more substantial lower-body posture adjustments with the salient target presented in the environment. Conclusions: Older adults with a low fear of falling can potentially benefit from a more salient target at their walking destination, whereas those with a high fear of falling were advised to implement a more straightforward falling intervention in their living areas.

## 1. Introduction

Falling is one of the leading causes of emergency room visits among the aging population; an estimated three million older adults per year require emergency medical attention [1]. Several widespread traumatic injuries, such as hip fractures, are usually directly caused by falling; for instance, falling accounts for over 95% of all hip fractures [2]. Fear of falling has been one key issue for older adults which prevents many from partaking in certain daily physical activities. Fear of falling has also been reported to increase the incidence of falling. It is reported that 25% of adults over the age of 75 had experienced falling during the past year, and over 41% have a fear of falling [3]. Of the people reporting fear of falling, 36% indicated they had suffered a falling accident during the past year [3]. Hence, there is an urgent need to understand how fear of falling acts on older adults’ independence and quality of life.

Most older adults highlight the notion of age in place in their homes [4] when considering independence and quality of life, especially during the pandemic period [5]. Individuals, as they age, tend to spend more time at home and in their communities due to reduced mobility and more substantial functional loss [6,7]. As a result, more than 60% of falls were reported to take place in older adults’ households [8,9], revealing the existence of certain environmental barriers in their homes. Their quality of life will therefore be negatively influenced by fallings and other safety issues, whether directly or indirectly, caused by these environmental barriers in their homes [10,11].

Poor visibility of the surroundings in homes is one of the key environmental barriers that contribute to falls and other safety issues [12,13]. Poor visibility issues include, but are not limited to, the poor design of a room (i.e., low contrast between target and surroundings), bad room arrangement (i.e., obstructed walkways), and inadequate ambient light. It has been demonstrated that visibility of the environment and the target is imperative for safe and effective locomotion in homes. For example, poor stepping accuracy and a slower gait speed were observed among older adults under restricted visual perception [14]. Moreover, an increase in gait variability and risk of contact has been associated with the decreased visibility of an obstacle [15].

Ensuring older adults’ visual perception of the surroundings, therefore, becomes vital to prevent falling injuries caused by poor visibility in the home. Previous research about visual input and older adults’ motor control concluded that visual perception of the surrounding becomes increasingly crucial for gait control as people age. Proactive visual planning is necessary to help older adults avoid collisions, as well as trips and falls [16]. Visual information is found to be especially critical in the final stages of older adults’ movement when precision control is needed [17,18]. In the case of walking, intermittent information is necessary for well-coordinated visual regulation over the last three strides as a target is approached [19].

There are studies incorporating indoor lighting assistance (e.g., front and side lighting) with the aim to help older adults with balance and postural control in their homes [20,21,22,23]. However, only a handful of the forms of lighting assistance were designed to improve the visibility of the target/destination. Furthermore, the majority of these studies focused primarily on the effects of lighting itself, neglecting to consider the fall concerns (i.e., fear of falling) and other mental and physical aspects of older adults. With that being said, there is a limited number of works in the literature aimed at investigating the impact of the visibility of the target on older adults’ walking destination and how that impact differs across older adults with different levels of fear of falling.

Therefore, the purpose of this study was to examine the effects of target salience on older adults’ gait and posture performance and investigate whether older adults with high fear of falling and low fear of falling change their behavior consistently or not. We hypothesized that older adults would improve their gait and posture with a more salient target at their walking destinations and older adults in the high-fear-of-falling group would be improved more obviously under the assistance of a more salient target at their destination.

## 2. Materials and Methods

### 2.1. Participants

This study enlisted the participation of fifteen older adults (165.5 ± 9.3 cm, nine females) from a local retirement community, all of whom were aged 75 years old and above. The sample size of this study was determined by relevant comparable studies from the literature [24,25], as well as practical constrictions (i.e., the resident population of the retirement community). Two main inclusion criteria were included during the participant recruitment phase: (1) age between 55 and 95 years old, and (2) ability to walk independently in a dimly lit environment without using walking assistive devices (e.g., wheelchair, and walker). Participants were excluded if they self-reported that (1) they had any trouble seeing, even with wearing glasses or contact lenses; or (2) they had a history of serious heart disease, serious stroke, or serious dementia. All participants can understand and speak English, so they had no difficulty conveying their feedback regarding the experiment. The University of Florida Institutional Review Board authorized the study (IRB Project #: IRB20190188).

### 2.2. Procedures

The experiment was conducted in participants’ bedrooms inside the retirement community. A total of eleven floor plans were included (two couples shared two floor types). Except for one floor type that had a turn in between, the path from the bedroom to the bathroom was mostly straight (6.90 ± 2.12 m) [26]. All participants have lived in their homes for an extended period of time and were familiar with the layout and furniture of their bedrooms. During the experiment setup phase, each participant was introduced to the study, signed the informed-consent form, and self-reported their fear of falling by filling out the Falls Efficacy Scale—International (FES-I) questionnaire [27]. The FES-I questionnaire, which contains 16 items/questions about participants’ concerns about falling, is a validated subjective evaluation method in research and clinical practice [28]. Meanwhile, researchers set up the lighting target (a LED strip from Commercial Electrics, Model # C624340), marked the testing locations, and simulated a dark environment (i.e., closing windows) in the bedrooms. By installing the LED strip and creating a dark environment, a salient target in the bedroom could be created (Figure 1). Researchers then attached a total of seven inertial measurement unit (IMU) sensors (MVN Awinda, Xsens Technologies BV, Enschede, The Netherlands) to participants’ lower bodies (Figure 2), specifically on the (1) pelvis, (2) left and right thigh, (3) left and right shank, and (4) left and right foot, as directed in the instruction manual [29]. These sensors were employed to record the motions of participants; therefore, necessary body dimension measurements and system calibration were also performed before the experiment.

During the experiment, participants were instructed to walk from the designated origin (i.e., near their beds) to the destination (i.e., near the bathroom entrance), with the lighting target turned on or off around the destination of the walking trials (Figure 1). Each participant completed four walking trials at their normal walking speed, two with the LED-lighted target (“salient” target) and the other two with the usual nightlight devices (“normal” target). During each trial, participants initiated the walking after hearing researchers’ verbal cues of “3-2-1-start” and terminated the walking once they reached the destination, the bathroom entrance. Participants were given ample time to rest between walking trials to avoid muscle fatigue and falling accidents. The LED strip was turned on by the researchers while participants were taking a break. The sequence of target salience conditions (salient vs. normal) was randomly presented across participants. After the walking trials, participants were assisted by researchers to remove the IMU sensors.

### 2.3. Experimental Design

To investigate the effect of target salience and fear of falling on older adults’ walking behavior, a mixed factorial design was utilized in this study, with the target salience (Target: salient vs. normal) as the within-subject variable and the fear of falling (FoF: low vs. high) as the between-subject factor. As mentioned in Section 2.2, under the salient target condition, participants walked toward the destination with the lighted LED strip, whereas, under the normal target condition, participants performed the same task with the LED strip turned off, but the usual nightlight turned on. In terms of the between-subject factor, using fear of falling (FoF) scores collected from the FES-I questionnaire, participants were classified into two FoF groups: low and high. Based on the cut-point developed by Delbaere and colleagues [28], the low-FoF group had the fear of falling scores, ranging from 16 to 22, whereas the high-FoF group had scores from 23 to 64. Participants who were classified into the low-FoF group were reported to have higher anxiety toward normal walking, as well as a lower perceived self-efficacy in preventing falls [30]. In this study, the gait pattern and lower-body posture were used to measure the motion response of participants when they were exposed to different target salience and fear of falling conditions. The details of the gait and posture variables are described in the next section.

### 2.4. Data Analysis

A standard gait analysis approach was performed in this study. Before gait detection and variable calculation, data exported from Xsens software (MVN Analyze 2019, Xsens Technologies BV, Enschede, Netherlands) were imported into MATLAB (R2020a, MathWorks, Natick, MA, USA). The following subsections detail the data preprocessing pipeline, which is derived from our previous publications [22,23,31].

#### 2.4.1. Gait Detection

Gait cycles must first be detected to calculate the gait and posture variables. A gait cycle was defined as the time between two heel strikes of the right leg (right heel strike, RHS), corresponding to 100% of the gait cycle (Figure 3). Heel strikes of the left leg (left heel strike, LHS), as well as toe offs (right and left toe offs, RTO and LTO), were also detected to define gait phases (e.g., double support phase) of a gait cycle (Figure 3). Data from sensors on both the left and right foot were used to detect gait cycles. Minimum values of the norm of foot resultant acceleration ($\underset{}{\min}\left(∥Acc_{\;foot}∥\right)$) were identified as heel strikes, whereas maximum values of the norm of the foot resultant acceleration ($\underset{}{\max}\left(∥Acc_{\;foot}∥\right)$) were used to find the toe offs [32]. The foot resultant acceleration was calculated from the x, y, and z components of both foot sensors after a second-order Butterworth filter (low-pass with 15 Hz cutoff frequency). After the gait detection, the timings (i.e., frame number) of each heel strike and toe off were stored to identify gait cycles and calculate gait and posture variables in each gait cycle.

#### 2.4.2. Variable Calculation

To compare the effect of target salience and fear of falling on walking behavior, variables characterizing gait pattern and lower-body posture were calculated for each gait cycle. Data of joint position, velocity, and acceleration, together with joint angles, were extracted and utilized for the calculation. A second-order Butterworth filter (lowpass with a cutoff frequency of 6 Hz) [33,34,35] was applied to the data to eliminate the impact of measurement noise.

A total of ten widely used spatiotemporal variables were chosen to describe gait patterns, including the average (i.e., mean) and variability (i.e., coefficient of variance) values of walking speed, stride length, stride width, stride time, and double support percentage. Here, in each gait cycle, the velocity data of the pelvic sensor were averaged to calculate walking speed, and the position data of the right and left foot sensors were used to compute the stride length and stride width [36]. The stride time was calculated from durations between consecutive right heel strikes in a gait cycle [37], whereas the double support percentage was based on the time intervals between heel strikes and the immediate toe offs on the opposite side in a gait cycle [38].

In addition to gait pattern variables, the sagittal flexion angles of bilateral hips, knees, and ankles at critical moments of a gait cycle were used to characterize the lower-body posture during walking [39,40]. Four categories of posture variables were included in the posture analysis (Figure 4): (1) the flexion angles of the right leg (i.e., hip, knee, and ankle) at RHS and the left leg at LHS were both extracted and combined to describe the posture of the moving leg at heel strikes, (2) the flexion angles of left leg at RHS and right leg at LHS were extracted and combined to describe the posture of the anchoring leg at heel strikes, (3) the flexion angles of the right leg at RTO and the left leg at LTO were extracted and combined to describe the posture of the moving leg at toe offs, and (4) the flexion angles of left leg at RTO and right leg at LTO were extracted and combined to describe the posture of the anchoring leg at toe offs.

### 2.5. Statistical Analysis

Two-way mixed analyses of variance (ANOVAs) were performed in R studio (Version 3.6.0) to compare the changes in gait pattern and lower-body posture in different conditions. Within-subject effects of the target salience condition (target: salient vs. normal), between-subject effects of the fear of falling (FoF: low vs. high), the interaction between two independent variables, and the random effect from participants were all included in the model. The *F*-value (F _(v1, v2)_), effect sizes ($\eta_{p}^{2}$), and *p*-value (*p*) were reported from ANOVAs. The indicative thresholds for effect sizes are set as small (0.01), medium (0.06), and large (0.14) [41]. If a significant interaction was found, paired t-tests were used for the follow-up comparison (salient/low vs. normal/low and salient/high vs. normal/high). The Shapiro–Wilk test and Levene’s test were used to verify the assumptions of normality and homogeneity of the ANOVA model residuals during the analysis, and the significance level (i.e., alpha level) was set at 0.05 for all ANOVAs and assumption tests.

## 3. Results

### 3.1. Characteristics of the Participants

The analyses included a total of sixty walking trials from fifteen participants. Participants were classified into two FoF groups, using scores from the FES-I questionnaire (refer to 2.3). There were seven participants in the low-FoF group (FES-I, 19.71 ± 1.25; stature, 161.49 ± 6.53 cm) and eight in the high-FoF group (FES-I, 30.75 ± 7.13; stature, 169.08 ± 10.36 cm). Between the two FoF groups, there were no significant differences in participants’ stature (*p* > 0.05) or shoe length (*p* > 0.05). Not surprisingly, the FES-I scores of participants in the low-FoF group were significantly lower than those in the high-FoF group (F_(1,13)_ = 16.20, $\eta_{p}^{2}$ = 0.55, and *p* = 0.001) (Table 1).

### 3.2. Gait Pattern—Spatial Variables

On average, regardless of the FoF groups, participants’ walking speed increased (F_(1,43)_ = 4.80, $\eta_{p}^{2}$ = 0.10, and *p* = 0.034) under the salient target condition (mean: 0.74 m/s), compared to the normal target condition (mean: 0.71 m/s) (Table 2 and Figure 5A). No significant differences were found in participants’ stride length and stride width, between two target conditions, and between two FoF groups (*p* > 0.05 for all).

### 3.3. Gait Pattern—Temporal Variables

Less double support variability was found (F _(1, 43)_ = 5.67, $\eta_{p}^{2}$ = 0.10, *p* = 0.022) when participants walked under the salient target condition (mean: 16.90), compared to the normal target condition (mean: 19.92), regardless of the FoF groups (Table 2 and Figure 5B). Other than the double support variability, there were no significant differences in the remaining temporal gait variables between the low- and high-FoF groups under two target conditions (*p* > 0.05 for all).

### 3.4. Hip Posture at Heel Strikes and Toe Offs

Significant differences were found for all hip posture variables. When the salient target was presented in the environment, participants’ moving leg tended to have less hip flexion during heel strikes (F _(1, 103)_ = 4.28, $\eta_{p}^{2}$ = 0.040, and *p* = 0.041) and toe offs (F _(1, 103)_ = 6.18, $\eta_{p}^{2}$ = 0.056, and *p* = 0.015), compared to the no-lighting condition (heel strike, 23.50° vs. 24.56°; toe off, 12.61° vs. 14.25°) (Table 2). Meanwhile, during the heel-strike phase, the hip flexion at the anchoring side decreased by 1.75° (F _(1, 103)_ = 7.54, $\eta_{p}^{2}$ = 0.068, *p* = 0.007) under the salient target condition (mean: −1.18°), compared to the normal target condition (mean: 0.57°) (Table 2). Furthermore, significant target x FoF interaction effects were found in the hip flexion at the moving leg during heel strikes (F _(1, 103)_ = 9.57, $\eta_{p}^{2}$ = 0.085, *p* = 0.003) (Figure 6) and at the anchoring leg during toe offs (F _(1, 103)_ = 6.76, $\eta_{p}^{2}$ = 0.061, *p* = 0.011) (Figure 7). A follow-up test showed that only the low-FoF group was observed to change their hip flexion angles at heel strikes and toe offs. At heel strikes, the hip flexion angle of their moving leg decreased under the salient target condition (25.36° vs. 22.44°, *p* = 0.001). Moreover, at toe offs, the hip flexion angle of their anchoring leg was reduced (*p* = 0.002) from 15.75° to 12.84° when the target was more salient in the environment.

## 4. Discussion

### 4.1. Summary

The aim of this study was to determine if a salient target improves the walking performance, more specifically the gait patterns and lower-body posture, of older adults with low and high fear of falling. The results showed that a more salient target at the destination led older adults to have a higher gait speed and improved stability, accompanied by less hip flexion during leveled walking. Our hypotheses were partially supported by the findings of the study. Older adults improved their gait stability and changed their hip flexion patterns with a more salient target presented at the walking destinations. However, contrary to our second hypothesis, which assumed the high-FoF group would benefit more from the salient target, older adults from the low-FoF group showed more obvious lower-body posture adjustments under the salient target condition.

### 4.2. Gait Pattern Changes Caused by a Salient Target

In line with our first hypothesis, a salient target induced older adults to adjust their gait patterns, as evidenced by changes in spatiotemporal variables. A higher gait speed and less double support variability were observed when participants walked with a more salient target. A four percent rise (from 0.71 m/s to 0.74 m/s) in gait speed indicated that participants had less hesitation when walking to the destination with a more salient target. This was consistent with our previous findings that less walking time was spent with destination-based visual cues in the dark environment [23]. Although the increased value in gait speed indicated less hesitation and more confidence among older adults when walking with a more salient target, our first hypnosis might still be challenged because there were studies suggesting that increased gait speed may increase the likelihood of a fall in older adults [42]. The reduction in double support variability of gait cycles then further resolved this issue and supported our first hypothesis. The double-support phase of a gait cycle was defined as the period where both the left and right feet are in contact with the ground, and it was used by people to restabilize themselves [43,44] and compensate for deviations in the center of mass [45] during walking. Because of that, the variability of double-support percentage was generally used as an indicator for walking stability [46,47]. Greater double-support variability is often associated with adverse outcomes, including increased risks of falls [46,48]. In our study, less double-support variability (19.92 vs. 16.90) was observed when participants walked under the salient target condition, indicating older adults’ better visuospatial function/ability (i.e., a person in relation to their environment) of their surroundings [47], leading to an improved gait control and mediated falling risks [49]. Cumulatively, the changes in the spatiotemporal variables showed a salient walking target helped older adults navigate more quickly in a relatively dark environment and stabilize their walking performance.

### 4.3. Lower-Body Posture Changes Caused by a Salient Target

Previous lighting studies in older adults mainly reported spatiotemporal comparisons between different lighting conditions. A limited body of the literature reports the posture adaptation caused by lights from different resources. Although our previous studies [23,50] compared joint kinematics (i.e., hips, knees, and ankles) between different lighting conditions, these studies only compared joint kinematics averaged out throughout a trial, which, in return, omitted the details from gait cycles and was not sensitive enough to capture differences between lighting conditions. The refined gait analysis from this study made an advancement to reveal participants’ lower-body posture adjustment between two target salience conditions. The results showed that, during heel strikes, participants’ moving legs showed a 1.06° reduction in hip flexion. Meanwhile, their hip flexion at the anchoring leg decreased by 1.75°. The trend of changes in flexion angle from both legs indicated a more anterior positioning of the pelvis under the salient target condition (Figure 8, left). It has been reported that the pelvis is normally treated as the approximation of the center of mass (COM) of a human [51,52]. The position and velocity of CoM determine how stable the gait is conducted [53]. Moreover, its relationship with the base of support (BOS) determines how well a human controls his/her walking and response to the stimulus from the environment [54]. Participants’ relatively anterior positioning of their COM (i.e., pelvis) could be explained by their voluntarily proactive responses to follow the direction of the changes of their BOS [54], as well as the direction between their body and the target. The same explanation also applies to the hip flexion reduction (14.25° vs. 12.61°) of the moving leg during the toe offs (Figure 8, right). Although the focus of this study is not on the inter-relationship between COM and BOS during different target salience, follow-up studies on COM–BOS interaction under different target salience in varied lighting conditions could be interesting and beneficial in terms of the design of nightlight devices.

### 4.4. Effects of the Fear of Falling on Gait and Posture

Older adults’ fear of falling is also an area of interest in our study. Fear of falling, as a validated approach to quantify the risk of falling, is always a popular research topic, whether it be any type of predictors or outcomes in falling-related studies. For example, one study reported an association between slower gait speed and stride length, as well as higher stride time variability, among individuals with a higher level of fear of falling [55]. In addition, fear of falling can also be cumulated, and its change has also been related to older age, lower dynamic balance, and poor visual contrast sensitivity [56]. In our study, we utilized fear of falling (FoF) as an indicator to classify the older adults’ anxiety toward walking and their perceived confidence in fall prevention. We hypothesized that older adults in the high-FoF group would benefit more with the assistance of a salient target at their destination. Findings of this study told a different story and indicated that, with a more salient target, older adults in the low-FoF group, rather than the high-FoF group, adjusted their postures with less sagittal flexion of hips and more anterior positioning of COM at heel strikes and toe offs. Given that older adults in the low-FoF group might have better body control and less functional loss [57,58], this finding was reasonable and suggested a stronger and more straightforward falling intervention (e.g., grabbing bar) to be implemented for those in the high-FoF group. Older adults who rated themselves in the high-FoF group were also advised to take extra precautions if they live independently, because their capacity to respond to the changes in the environment and regulate the body’s momentum might be affected to a certain degree.

### 4.5. Limitations and Future Directions

Limitations of the study include a relatively small sample size and a lack of representation of older adults with some disability. A larger sample size might have increased the statistical power so that other gait and posture variables, such as stride length and hip flexion of the anchoring leg at toe offs, may have the potential to reach statistical significance. Other than the variables reported in the Results Section, variables depicting the variability of the spatial aspect of gait, as well as the joint kinematics of knees and ankles, were also extracted and compared (Appendix A). Although not many conclusive findings can be summarized in this study based on the variables in Appendix A, a more diverse and inclusive sampling of participants would make the story different. Participants who can walk independently without assistive devices (e.g., walkers and canes) were the target of this study. However, there is a certain number of older adults that live independently and with some types of physical impairment. It is worthwhile to take this into consideration and develop lighting intervention for older adults with different functional abilities. Therefore, an expansion of the sample size and sample characteristics is one of our future directions. Aside from the limitation in participant sampling, there exist some constraints in the features of lighting, as well. Other features of the salient target (e.g., the illuminance level and ignition duration) may also influence older adults’ behavior and satisfaction in the dark environment. Future studies could extend the research direction and investigate whether older adults are affected by other aspects of the salient target. Another future direction for this study, based on the discussion in Section 4.3, will be further investigating posture control, specifically the COM–BOS interaction, under different target salience in varied lighting conditions.

## 5. Conclusions

A more salient target at the destination increased older adults’ motion speed, improved their gait stability, and changed their hip flexion patterns when walking in a dark environment. Older adults with a low fear of falling were observed to benefit more from the salient target. Older adults with a high fear of falling were advised to implement a more straightforward falling intervention in their living areas. Future research could investigate the COM–BOS interaction under a different target salience and different lighting conditions.

## Figures and Tables

**Figure 1 sensors-22-09352-f001:**
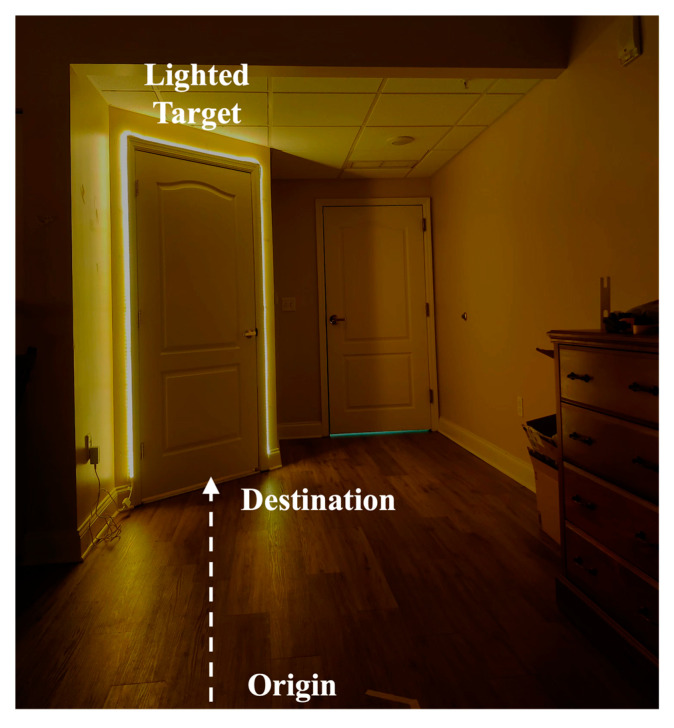
A conceptual illustration (rather than an exact replication due to participation privacy concerns) of the salient target by installing the LED strip in a dark environment.

**Figure 2 sensors-22-09352-f002:**
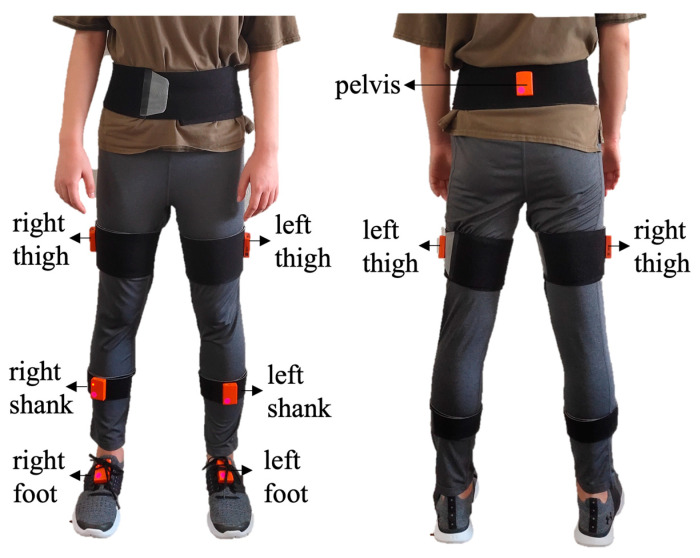
Seven IMU sensors were attached to the lower body.

**Figure 3 sensors-22-09352-f003:**
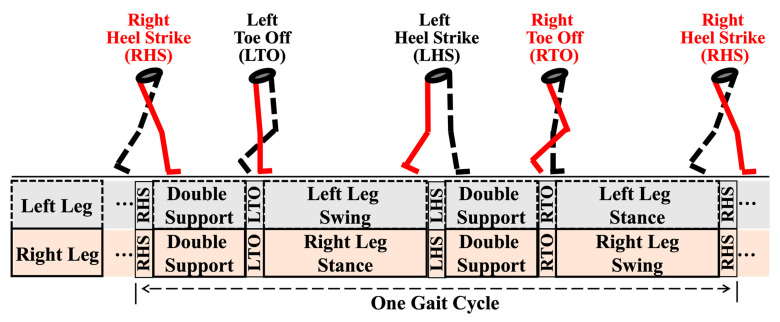
Critical moments (RHS, LTO, LHS, and RTO) and phases (double support, swing, and single leg stance) of one gait cycle.

**Figure 4 sensors-22-09352-f004:**
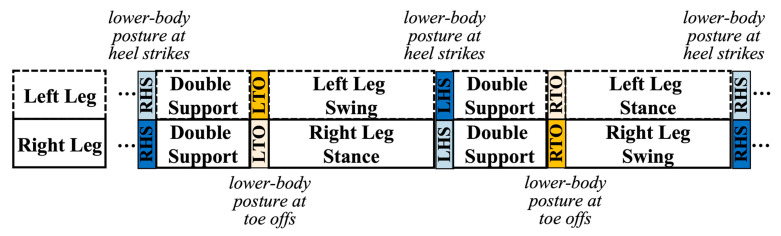
The posture of the moving leg at heel strikes (dark blue), of the anchoring leg at heel strikes (light blue), of the moving leg at toe offs (dark yellow), and of the anchoring leg at toe offs (light yellow).

**Figure 5 sensors-22-09352-f005:**
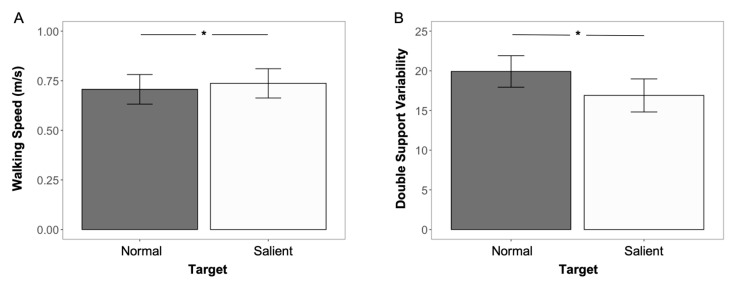
Comparison of spatiotemporal variables under two target conditions: significant difference in (**A**) walking speed and (**B**) double support variability between normal and salient conditions. The asterisk symbol (*) was used to indicate a significant difference with an alpha level of 0.05.

**Figure 6 sensors-22-09352-f006:**
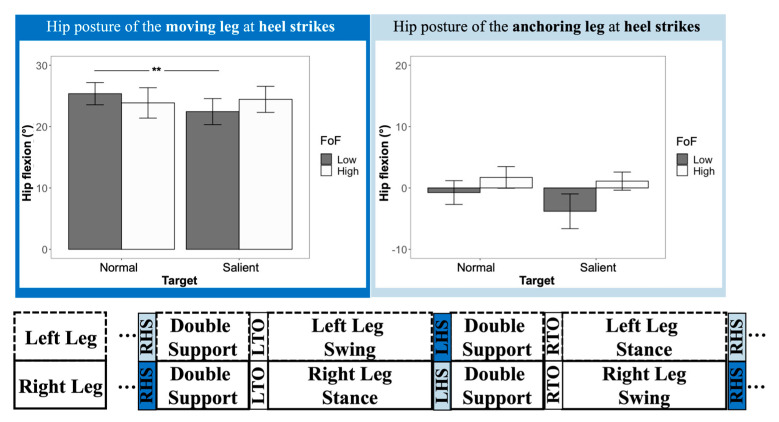
Hip flexion of the moving leg (dark blue) and the anchoring leg (light blue) at heel strikes. The double-asterisk symbol (**) was used to indicate a significant difference with an alpha level of 0.01.

**Figure 7 sensors-22-09352-f007:**
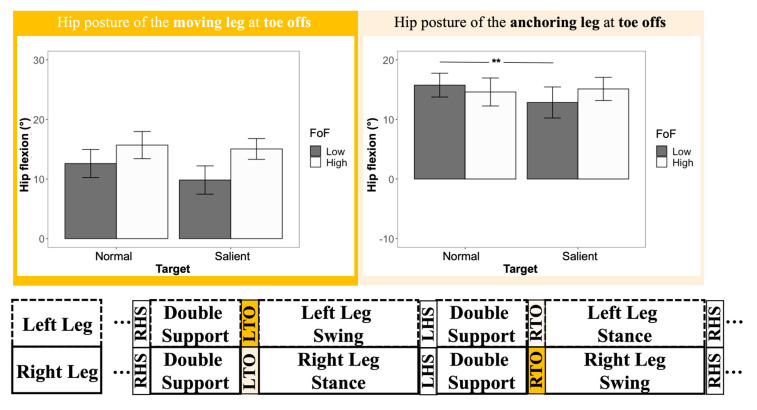
Hip flexion of the moving leg (dark yellow) and the anchoring leg (light yellow) at toe offs. The double-asterisk symbol (**) was used to indicate a significant difference with an alpha level of 0.01.

**Figure 8 sensors-22-09352-f008:**
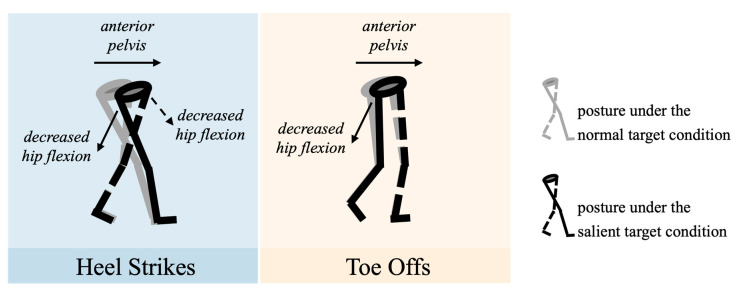
Illustration of participants’ lower-body posture at heel strikes (**left**) and toe offs (**right**) under the normal target condition (gray) and the salient target condition (black).

**Table 1 sensors-22-09352-t001:** Summary of fifteen participants’ characteristics—mean (SD) or N (%) within the FoF group.

Characteristics	FoF		
	Low	High	*p*-Value
Numbers (Total)	7	8	-
FES-I Score	19.71 (1.25)	30.75 (7.13)	0.001
Stature (cm)	161.49 (6.53)	169.08 (10.36)	0.120
Shoe Length (cm)	27.96 (1.85)	29.54 (1.88)	0.126

**Table 2 sensors-22-09352-t002:** Mean, standard deviation (SD) values, and ANOVA results of gait variables.

Type		Variables	Target			FoF			Target × FoF
			Normal	Salient	*p*-Value	Low	High	*p*-Value	*p*-Value
Gait Pattern—Spatial		Walking Speed (m/s)	0.71 (0.20)	0.74 (0.20)	0.034	0.81 (0.19)	0.64 (0.18)	0.104	0.430
		Stride Length (m)	0.82 (0.22)	0.85 (0.22)	0.091	0.91 (0.22)	0.77 (0.20)	0.239	0.422
		Stride Width (m)	0.21 (0.03)	0.21 (0.04)	0.130	0.20 (0.03)	0.22 (0.04)	0.226	0.205
Gait Pattern—Temporal		Stride Time (s)	1.24 (0.15)	1.24 (0.14)	0.845	1.18 (0.11)	1.29 (0.15)	0.114	0.841
		Double Support (%)	37.86 (7.89)	36.75 (6.85)	0.285	35.56 (7.90)	38.84 (6.56)	0.366	0.370
		Stride Time Variability	8.62 (3.64)	7.85 (3.74)	0.403	7.74 (3.62)	8.66 (3.74)	0.486	0.230
		Double Support Variability	19.92 (5.44)	16.90 (5.70)	0.022	19.94 (5.90)	17.07 (5.31)	0.138	0.952
Hip Posture	Heel Strike	Moving Leg (°)	24.56 (6.07)	23.50 (5.86)	0.041	23.90 (5.37)	24.15 (6.48)	0.932	0.003
		Anchoring Leg (°)	0.57 (5.16)	−1.18 (6.40)	0.007	−2.27 (6.54)	1.42 (4.57)	0.142	0.069
	Toe Off	Moving Leg (°)	14.25 (6.48)	12.61 (6.14)	0.015	11.21 (6.35)	15.37 (5.70)	0.139	0.127
		Anchoring Leg (°)	15.14 (6.02)	14.05 (6.24)	0.071	14.30 (6.26)	14.86 (6.06)	0.844	0.011

## Data Availability

The data presented in this study are available upon request from the corresponding author.

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
