# Peer review of "Salient Targets and Fear of Falling Changed the Gait Pattern and Joint Kinematic of Older Adults"

_sensors, 2022, doi:10.3390/s22239352_

Round 1
Reviewer 1 Report
The paper delivers a clear message to readers about the influence of fear of falling on the style of walking depending on lite conditions.
Methodology is clearly described with details regarding gait pattern and related fear characteristics. Authors used proper statistical methods while analyzing measured data.
The presented experimental research has also some limitations that were partially described by authors and promised to improve during future experimental work. The light on the way to bathroom is definitely a useful service for seniors. However, results and the satisfaction of users may be affected by many factors, some of them are mentioned by authors, but there are more, for example when is the light initiated (it may be disturbing to be on for the whole night), etc. Just to summarize, the paper has good scientific quality.
References (54) are relevant, mostly dated after 2010, just I have found not sufficient data for ref 26, it requires more data to identify the source.
Author Response
Thank you for your valuable feedback. We appreciate your comments about the study limitations and reference materials. We have made the following changes to the manuscript in response to your suggestions:
- We have included the time when the light was turned on during the experiment (Lines 136-137), and it reads “The LED strip was turned on by the researchers while participants were taking a break”. We have also added one limitation in terms of the light initiation time on users’ satisfaction. The corresponding paragraph sits in Lines 377 - 381, and it reads “Aside from the limitation in participant sampling, there exist some constraints in the features of lighting as well. Other features of the salient target (e.g., the illuminance level and ignition duration) may also influence older adults’ behavior and satisfaction in the dark environment. Future studies could extend the research direction and investigate whether older adults are affected by other aspects of the salient target.
- We have added one more reference (reference 58) in addition to the original reference (54) to support our statement “Given that older adults in the low FoF group might have better body control and less functional loss”. And we have reformatted reference (26) (now reference 29) and added doi to it to make it more identifiable.
References added:
29. Schepers, M.; Giuberti, M.; Bellusci, G. Xsens MVN: Consistent Tracking of Human Motion Using Inertial Sensing. Xsens Technol 2018, 8, doi:10.13140/RG.2.2.22099.07205.
58. Bjerk, M.; Brovold, T.; Skelton, D.A.; Bergland, A. Associations between Health-Related Quality of Life, Physical Function and Fear of Falling in Older Fallers Receiving Home Care. BMC Geriatr 2018, 18, 253, doi:10.1186/s12877-018-0945-6.
Reviewer 2 Report
How did you determine the sample size?
'they had any trouble seeing, even with wearing glasses or contact'--did you assess this using a test? If yes, which test?
'they had a history of serious heart disease, stroke, or dementia'--please write this a bit clearer. Do you mean serious stroke or any stroke? Serious dementia or any dementia? Did you do any cognitive tests to check whether they had dementia which had not been diagnosed?
'The experiment was conducted in participants’ bedrooms inside the retirement community'--Were all bedrooms the same? Same lighting, same shape, same design, same type and number of furniture in the same position?
Did you exclude people who did not understand English?
Did you exclude people with movement disorders e.g. ataxia, chorea, tremor, Tourette's syndrome, or vertigo?
Were all participants familiar with the room they were staying in? For example, was there anyone who had just recently been staying in the room (e.g. moved in the previous day)?
Did you exclude people with problems in spatial memory or navigation e.g. developmental topographical disorientation?
MINOR
It would be good to include dois to all references
Author Response
How did you determine the sample size?
Response: Thank you for your question. We have described our strategy to determine the sample size in the manuscript (Lines 93 - 95) and it reads “The sample size of this study was determined by relevant comparable studies from the literature [24,25] as well as practical constrictions (i.e., the resident population of the retirement community).”
'they had any trouble seeing, even with wearing glasses or contact'--did you assess this using a test? If yes, which test?
Response: Thank you. The assessment was based on the demographic questionnaire collected from participants. Therefore, instead of any tests, self-reported responses to the corresponding questions were utilized to exclude participants. We have indicated that in the method section (Line 98), and now it reads “Participants were excluded if they self-reported that: 1) they had any trouble seeing…”
'they had a history of serious heart disease, stroke, or dementia'--please write this a bit clearer. Do you mean serious stroke or any stroke? Serious dementia or any dementia? Did you do any cognitive tests to check whether they had dementia which had not been diagnosed?
Response: Thank you for the questions. By referring to stroke and dementia, we are talking about serious stroke and serious dementia. And we have explicitly indicated that in the manuscript (Line 100). We did not conduct cognitive tests during the experiment because we 1) relied on the self-reported answers from our participants and 2) intended to minimize the experiment duration in order to better mitigate fatigue and keep our participants engaged.
'The experiment was conducted in participants’ bedrooms inside the retirement community'--Were all bedrooms the same? Same lighting, same shape, same design, same type and number of furniture in the same position?
Response: Thank you for pointing this out. In short, the bedroom layouts inside the retirement community are not the same. There are different floor plans in the retirement community, and we intended not to control the bedroom layout because we consider that testing in participants' own homes increase the ecological validity of the paradigm. To make this clear, we have included one paragraph describing the settings of the bedrooms in the study (Lines 106 - 108):
“A total of eleven floor plans were included (two couples shared two floor types). Except for one floor type that had a turn in between, the path from the bedroom to the bathroom was mostly straight (6.90 ± 2.12 m).”
Did you exclude people who did not understand English?
Response: Thank you. All of the participants who contacted us could understand and speak English although we did not exclude any non-English speaking participants. We had Spanish speaking team members (Latinos represent a large portion of the local population), but just ended up not having any recruited. We have also included this information in the method section (Lines 100 - 102): “All participants can understand and speak English so they had no difficulty conveying their feedback regarding the experiment.”
Did you exclude people with movement disorders e.g. ataxia, chorea, tremor, Tourette's syndrome, or vertigo?
Response: Thank you. We did not exclude people with movement disorders (ataxia, chorea, tremor, Tourette's syndrome, or vertigo). The criteria for us to include participants was indicated in 2.1., that participants who are able to walk independently in a dimly lit environment without using walking assistive devices were recruited.
Were all participants familiar with the room they were staying in? For example, was there anyone who had just recently been staying in the room (e.g. moved in the previous day)?
Response: Thank you for your inquiries. All participants, to the best of the authors' knowledge, had lived in their homes for an extended period of time and were familiar with the layout and furniture of their bedrooms. And we mentioned that in the manuscript (Lines 108 - 110).
Did you exclude people with problems in spatial memory or navigation e.g. developmental topographical disorientation?
Response: Thank you. We did not disqualify anyone who had difficulty with spatial memory or navigation. And to the best of the authors' knowledge, no participants reported experiencing spatial memory or navigation problems.
MINOR
It would be good to include dois to all references
Response: Thank you for your suggestions. We have included all dois to the journal articles and conference proceedings. Book references were not assigned by a DOI, however, we have double-checked the format of the book references to make sure it is standard and contains enough data to identify the source.
Round 2
Reviewer 2 Report
My comments and questions have now been addressed by the authors.